# User Experience Design for Social Robots: A Case Study in Integrating Embodiment

**DOI:** 10.3390/s23115274

**Published:** 2023-06-01

**Authors:** Ana Corrales-Paredes, Diego Ortega Sanz, María-José Terrón-López, Verónica Egido-García

**Affiliations:** 1Science, Computation and Technology Department, School of Architecture, Engineering and Design, Universidad Europea de Madrid, 28670 Villaviciosa de Odón, Spain; anadelvalle.corrales@universidadeuropea.es; 2Aerospace and Industrial Engineering Department, School of Architecture, Engineering and Design, Universidad Europea de Madrid, 28670 Villaviciosa de Odón, Spain; diego.ortega@universidadeuropea.es; 3Vice-Dean Engineering, School of Architecture, Engineering and Design, Universidad Europea de Madrid, 28670 Villaviciosa de Odón, Spain

**Keywords:** social robots, human–robot interaction, user experience, embodiment

## Abstract

Social robotics is an emerging field with a high level of innovation. For many years, it was a concept framed in the literature and theoretical approaches. Scientific and technological advances have made it possible for robots to progressively make their way into different areas of our society, and now, they are ready to make the leap out of the industry and extend their presence into our daily lives. In this sense, user experience plays a fundamental role in achieving a smooth and natural interaction between robots and humans. This research focused on the user experience approach in terms of the embodiment of a robot, centring on its movements, gestures, and dialogues. The aim was to investigate how the interaction between robotic platforms and humans takes place and what differential aspects should be considered when designing the robot tasks. To achieve this objective, a qualitative and quantitative study was conducted based on a real interview between several human users and the robotic platform. The data were gathered by recording the session and having each user complete a form. The results showed that participants generally enjoyed interacting with the robot and found it engaging, which led to greater trust and satisfaction. However, delays and errors in the robot’s responses caused frustration and disconnection. The study found that incorporating embodiment into the design of the robot improved the user experience, and the robot’s personality and behaviour were significant factors. It was concluded that robotic platforms and their appearance, movements, and way of communicating have a decisive influence on the user’s opinion and the way they interact.

## 1. Introduction

In recent years, robotics has been expanding its field of action, from households [1] to robots aimed at facilitating and improving the quality of life of people [2]. More recently, the COVID-19 pandemic has led to a substantial increase in studies on social robotics [3]. The effectiveness of Human–Robot Interaction (HRI) in social robotics is a big challenge. Research in social robotics is focused on many applications [4], for example in education [5,6], as well as works referring to people with cognitive problems, or special needs, or the elderly [7,8,9,10,11,12,13,14], localisation and navigation—giving or asking help to reach a destination—[15], among other various purposes, where a user plays a primary role.

User Experience (from now on UX) is a key concept for the design of physical and digital products, such as devices, software, web pages, and mobile apps. According to ISO 9241-210 [16], UX is defined as: “A person’s perceptions and responses that result from the use and/or anticipated use of a product, system or service”. Some authors define UX as the feelings that the user has when using a product [17] or the quality of experience during the interaction [18,19]. Several aspects are important for the user to have a good interaction experience, such as acceptance, usability, ease of learning, security, trust, and credibility [20].

In the field of social robotics, considering the user’s feelings when interacting with the robot could give us the keys to designing better user-centric applications. Several researchers have examined the role and importance of UX for socially interactive robots [21,22]; nevertheless, the interaction design principles are not only relevant for evaluating social robots, but can also be applied to enhance collaborative robotics systems [23], the structure of human–robot dialogue [24], or thinking about the design of the applications [25,26].

Researchers have studied the importance of evaluation techniques or methodologies. Apraiz et al. [27] identified methodologies that evaluate the Human–Robot Interaction (HRI) from a human-centred approach, and their results showed the importance of considering different types of measurements: qualitative and quantitative; objective and subjective. Shourmasti et al. [28] carried out a systematic literature review that relied on the PRISMA guidelines, and their findings indicated that the most-common methods used to evaluate UX in social robots are questionnaires and interviews and that UX evaluations can provide early feedback, allowing developers to identify and address issues at an early stage in the development process. Lindblom et al. [29] emphasised the significance of using various methods to strengthen the insights derived from UX evaluation, and they enhanced the importance of qualitative data in this type of evaluation. Furthermore, they presented the development of a systematic user experience evaluation framework called ANEMONE (action and intention recognition in human–robot interaction), and the framework was designed to measure and assess the mutual recognition of actions and intentions between humans and robots, which is a key factor in enabling high-quality interaction between the two [30].

Users’ likes and dislikes will make the human–robot interaction more meaningful, where usability and interaction experience will be the main guidelines that mark the design process. According to Shamonsky [31], UX in a robot requires thinking about the following design topics: the context of use, safety, physical design and ergonomics, general interaction, interaction modalities, Artificial Intelligence (AI), agency, and autonomy. This article focused on the user experience approach in terms of the embodiment of the robot.

Barsalou et al. referred to social embodiment as the “states of the body”, which play a fundamental role during the process of social interaction, including postures, movements, and facial expressions; these bodily states themselves produce affective states [32]. The interest in the term embodiment is beginning to take on special relevance in areas such as robotics and interaction in digital environments. Recent research analyses the impact of embodiment to achieve naturalness in nonverbal communication between humans, social robots, and virtual agents [33,34,35]. Deng et al. gave a comprehensive state-of-the-art analysis of the role of physical embodiment in social robotics [36]. They suggested that the way a robot embodies itself can have a significantly impact on how it performs and how people perceive its social interactions. This highlights the importance of considering various parameters when designing both the robot and its interaction environment.

Natural human–robot interaction is essential in social robotics [37] and key to obtaining a good communication and user experience. Thus far, humanoid-looking robots have generally been used as physical agents to establish communication with the user. However, despite the efforts to make a robot as similar to a human as possible, factors such as the speed of movement, or the dissociation of movement, or the dissociation of voice and body language, among others, make that a part that our brain detects and categorises as not real [38,39,40].

The discrepancy between a robot’s non-natural movements and its appearance can sometimes lead to rejection or a lack of empathy from users, often subconsciously [41]. This is partly due to technical limitations and the materials used in the robot’s construction. As a result, there has been an increase in research on soft robotics, which aims to replicate human and animal movement characteristics [42].

However, research about the positive effects of the presence of social robotics on people through behaviours that increase the degree of empathy, collaboration, and trust can be found in several articles and by several authors [43,44]. Some comparative studies showed a higher degree of satisfaction in cases where the robot was physically present instead of a digital avatar when studying both body movements [41,45]. According to recent research, the embodiment resulting from multi-sensory bodily interactions among individuals has been found to enhance social attitudes, including closeness and empathy, while also reducing racial biases [46,47]. More recently, Zamboni et al. proposed that the semblance of life in non-living devices comes from four communicative elements: body, behaviour, setting, and the machine’s name [48].

Designing that autonomy in UX terms will make robotic applications more user-friendly, avoiding user exhaustion or boredom during interaction [49,50]; that is, it would increase their usability [51].

On this basis, the research question that guides our work is the following: How does a person feel about a robotic platform in a communication process?

Answering this question will allow us to know and understand in depth the impact of the embodiment of a robot in the interaction with people. This perspective pivots on how embodiment affects users during their user experience interacting with a robot [52,53,54].

The article focuses on examining the user experience in relation to the embodiment of the robot and verbal communication. This encompasses analysing how the robot’s physical appearance and movement can influence the user’s perception and overall interaction experience. Additionally, designers should consider incorporating gestures, emotional expressions, and dialogue to create a more engaging and relatable robot.

To explore this topic further, our study will address the following exploratory questions:How does a robot’s verbal and nonverbal communication impact the user experience?When communicating with a robot, how do users perceive its nature and to what extent do they view it as a machine or a human-like entity?

The structure of the paper is as follows. First, the methodology followed will be presented and the design of the experiment and how the intervention was conducted will be described. Section 3 gives the preliminary results based on the participants and the researchers’ perceptions. Finally, conclusions and future works about the perception of people in a communication process with a robot are provided.

## 2. Methodology

According to ISO 9241-210:2019 [16], human-centred design involves understanding the needs of the users, involving them in the design process, and prioritising their feedback. Thus, the research emphasises the importance of involving potential users in the design process to ensure that the resulting solution addresses their needs and preferences. Therefore, to answer the exploratory questions posed, the current study adopted an exploratory approach, which aimed to gain a deeper understanding of the existing problem.

While the exploratory approach does not yield conclusive results, it allows for a contextualised understanding of the relevant behaviours and perceptions of the study participants without requiring a large sample size. The goal is to investigate the phenomenon within its natural setting and without manipulating the behaviours of the participants. By adopting this framework, the study strove to unveil valuable insights into the issues under investigation [55,56]. In this mode of inquiry, small sample sizes are favoured to facilitate a case-oriented analysis, which forms the core of the investigation.

The design approach used in this research was determined by the information acquired from the potential users. This initial phase involved recruiting participants from various groups, including university students, university employees, and parents, among others. To accomplish this, social media platforms and the virtual campus were used as effective channels for participant recruitment. These avenues allowed us to reach a wide range of potential participants and ensure a diverse representation within the study. By seeking input from a diverse range of participants, the research aimed to gather valuable and unbiased data. It is important to note that participation in the study was completely voluntary.

To gather information on the user experience, the research employed both quantitative and qualitative methods using a questionnaire. Multiple data sources were gathered and triangulated to give validity to the study: a questionnaire, the robot technician’s personal diaries of each session, and the video recording of the sessions. By employing this mixed-methods approach, a more thorough comprehension of the user experience can be achieved, offering valuable insights into participants’ attitudes and preferences towards the robot-assisted platform.

The questionnaire was designed to elicit users’ preferences for specific aspects of the robot and its conversational form, aligned with the research objectives. In addition to incorporating closed-ended questions, the questionnaire included open-ended questions to encourage participants to express their opinions and contribute insights [57].

Drawing inspiration from the widely recognised Godspeed questionnaire, a renowned tool for assessing users’ direct impressions of their interactions with both robots and humans, the questionnaire was designed [58,59]. By utilising this established questionnaire as a foundation, the research benefited from established metrics and methodologies for evaluating users’ experiences with the robot-assisted platform. Furthermore, employing a well-established questionnaire facilitated comparisons with other studies and enhanced the reliability and validity of the results. The questionnaire was developed in Spanish using MS Forms, which streamlined the process of collecting responses from participants and facilitated the analysis and exportation of the results.

The questionnaire administered to the participants was divided into three sections. The first section aimed to gather contextual information about each respondent. The second section comprised closed-ended questions, which used a 5-point Likert scale ranging from “A lot” to “Nothing” to evaluate specific aspects of the robot’s communication. Finally, the third section consisted of open-ended questions designed to retrieve qualitative information about the participants’ perceptions.

Finally, the qualitative data, consisting of open-ended questions, video recordings, and the technician’s diary, were collected with the intention of conducting a content analysis using coding and interpretative analysis techniques.

The methodology used in this study, employing a mixed-methods approach, is depicted in Figure 1. This approach involved analysing the problem, designing interviews and questionnaire questions, as well as experimenting with user interactions. By using this approach, both quantitative and qualitative data were gathered, which allowed for a more comprehensive understanding of the robot’s communication with individuals. It provided valuable insights into various aspects of the interaction, allowing for a more comprehensive evaluation of the overall performance.

### 2.1. Experiment Design

To gather the information needed, the humanoid robot Pepper (Softbank Robotics, Tokyo, Japan) was used. This robotic platform has been designed to interact with people; its flexible movements and ability to move were fundamental to carrying out the study [60], as it is a robot capable of expression. However, the Pepper platform does not develop new strategies on its own, but its decisions come from the previously programmed strategy, which is why the collaboration of researchers from other areas of knowledge is necessary. The study helped us investigate how people interpret the embodiment of Pepper [61,62]. Pepper uses head movements, gestures, and postures, coloured eyes, sounds, and, in some cases, a dynamic representation of emotions through approaches and/or distances from the interlocutor, and the study was conducted to explore how people interact with it.

The robot’s features and external design were to emulate a human-like drawing without trying to give realism to a human figure in order to minimise the possible aversion or fear of interacting with it. The robot is 1.60 metres tall and has no legs, and its arms, head, and torso are mobile, allowing it to interact with the environment and users. The robot also has cameras and sensors to capture information from the environment.

Different parameters have been programmed into the robot, which remained stable during the experiment:Voice and sound: The voice of the robot is female, clearly distinguishable from a human voice. The tone is stable during the conversation, with slight variations depending on the mood of the responses (sadness, happiness, confusion, etc.).Personality: The personality of the robot is extroverted, takes initiative, and actively questions the user. The robot is able to adapt its behaviour and the conversation according to the user’s behaviour, detecting if he/she is shy, if he/she feels like talking, if he/she is scared, etc., generating an individual profile for each user.

To evaluate the effectiveness of the robot’s communication with people, a lab-based user study was carried out. The study was conducted in the Robot Learning Lab, a room measuring 5 m × 3 m and divided into two spaces by a screen (Figure 2 and Figure 3). One space was occupied by the robot technician, who monitored the experiment and made real-time adjustments as needed. In another space of the room, the Pepper robot was positioned next to a chair where the participant was invited to take a seat and start the interaction.

Participants were recruited through various means, including social media, the online campus, and posters on campus. Informed consent is a fundamental ethical aspect of research involving human subjects. It ensures that the participants understand the nature and purpose of the study, the potential risks and benefits, and their rights as research subjects. Therefore, all participants were required to sign an “Information Sheet and Informed Consent” form to protect their confidentiality and privacy and to ensure that the study adhered to ethical guidelines. At this moment, researchers provided adequate information to participants and answered any questions they had to ensure that they fully comprehended the study and made an informed decision on whether or not to participate.

**Figure 3 sensors-23-05274-f003:**
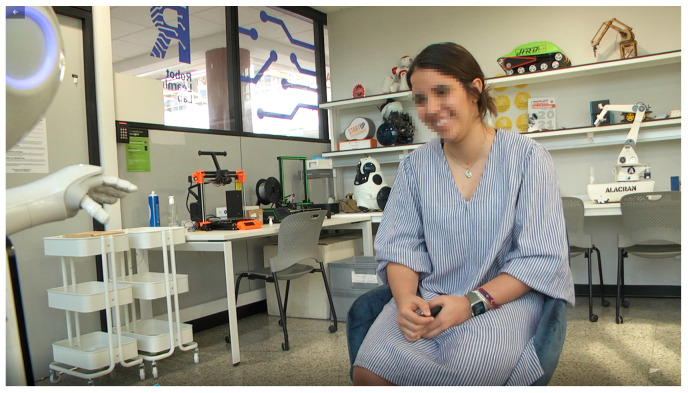
User interacting with Pepper.

Figure 2 depicts the layout of the experimental environment. A video camera was installed behind Pepper to capture the participants’ reactions during the session for further analysis.

Figure 3 shows an extract of the videos recorded during the sessions. A camera was fixed in a position that allowed the user to focus on the robot, using an angle that would not distract the participants and allow them to concentrate on the robot. The aim of the recordings was to obtain nonverbal information from the users and to observe their interaction with the robot.

Prior to the sessions, the technician entered the Robot Learning Lab to ensure that Pepper was functioning properly and to avoid any disruptions during the study. Each session lasted between 1 and 5 min, depending on the participants’ reactions.

A total of 36 participants (12 men and 24 women) attended the study throughout the day. They entered the lab one by one and were asked to fill out the questionnaire on a tablet after completing the session with Pepper. The questionnaire was divided into three sections, as previously mentioned.

After each session, the evaluator recorded a personal diary of what occurred during that session. This allowed for additional insights and observations to be included in the analysis of the study.

### 2.2. Session Structure: Interaction with Pepper

The aim was to create a kind and non-aggressive interaction between the participants and Pepper, not only through verbal communication, but also through the robot’s embodiment, such as movements and coloured eyes. To meet this requirement, the architecture of Pepper was carefully designed to analyse participants’ responses during the interaction.

Choreographe version 2.8.7.4, the robot programming tool based on the programming language C++ and Python were used for the development and behavioural programming of the robot. This tool allowed us to create animations and behaviours for Pepper and test them in a simulated environment before conducting the study with real users.

Figure 4 presents a diagram of the conversation that was programmed for the study.

Upon entering the lab, the first interaction with the participants always began with a greeting script, where Pepper would initiate a brief conversation by saying “Hello”. This script was designed to establish emotional interaction with the participant. The use of emotional interaction can help create a more engaging and positive user experience. Pepper would then ask for the participant’s name and occupation, and if the participant was a student, it would inquire about his/her field of study. The dialogue varied depending on the responses given by the participant. Pepper always finished the interaction with a direct question. By ending the initial interaction with a direct question, the robot can transition to a different script and move the conversation forward.

This approach can help keep the participant engaged and interested in the interaction with the robot. Pepper’s responses were programmed to include movements and sounds, such as raising its arms, to convey behaviour approval and positive reinforcement. The robot’s programming followed a structure that allowed it to adapt to different responses and initiate different conversations with the user. Figure 5 showcases the decision-making process and responses provided by the robot, thereby presenting a comprehensive scheme of the conversation structure.

The answers of “Pepper” sometimes include sounds and movements such as raising the arms or conveying different forms of feedback to reflect behaviour approval and positive reinforcement. Furthermore, Pepper asks the participant to interact with it.

The movements, gestures, and sounds were studied and programmed for each phrase with careful consideration to give the robot maximum expressivity. There were expressions of euphoria, sadness, laughter in jokes, positions that denote interest or active listening, etc. For this reason, a table was designed to gather information about each gesture, which included an identification name and a description of what the movement involves to convey the emotion it is related to. These descriptions included the motion of each part of the robot during the gesture and the colour of the eyes in case they changed from the default white colour. For example, the gesture with the label “ScratchEye_1” consists of lowering the head while taking the back part of the wrist as if it were scratching its eye after crying to convey sadness. An extract of the table designed is shown in Table 1, which includes the gestures used during the interaction.

## 3. Results and Discussions

To analyse the response of participants to the interaction with Pepper, multiple data were gathered, including questionnaires with Likert-scale answers, four open-ended questions, video recordings of each session, and the programmer’s personal diary of the session. Each session lasted no longer than five minutes.

The open-ended questions, programmer diary, and videos were analysed qualitatively using thematic coding and interpretative techniques in NVivo 13 to identify patterns and themes in the data [63,64,65].

To ensure consistency in coding, successive meetings were held to establish coding dynamics, and the videos were reviewed by all researchers, with observational categories determined through consensus and discrepancies discussed.

The analysis of the open-ended answers and programmer notes involved identifying content units and establishing initial categories, followed by a saturation of the information. To clarify and triangulate the data, several important quotes from the participants were also extracted. The researchers convened to engage in an in-depth interpretation and correlation of the outcomes obtained from both the quantitative and qualitative analyses. Finally, a cohesive narrative was crafted to establish meaningful connections among the findings.

### 3.1. Quantitative Results

At the end of the session, participants were given an electronic form to complete, which consisted of three groups of questions. The first group included general questions about the participant’s gender, age, and education level, which were used to differentiate between different population groups. The second group of questions focused on the participant’s perception of the robot, including its expressions, interactions, and areas for improvement in the conversation. The final group of questions aimed to understand the participant’s emotions during the session, such as their feelings towards the robot, empathy, and closeness. All participants completed the survey, including individuals with diverse characteristics, which were filtered by gender, age, and knowledge area.

The study had a total of 36 participants, consisting of 12 men and 24 women. The majority of participants (80.56%) were students, and as indicated in Table 2, most of them were under the age of 25. The analysis of the questionnaire results was performed by simultaneously observing the recorded videos during the session and triangulating the results. This was very important as the recordings provided us with relevant information about the participants, not only in terms of their attitudes towards the robot, but also the differences shown based on their age and gender.

The questions were analysed by grouping them as shown in Figure 6.

Users’ perceptions of the robot were explored, including whether they considered it to be fake, human, a living being, or a machine. Overall, users had positive perceptions of the robot. To analyse the results, the answers found were cross-referenced with the analysed videos of each participant.

All the participants over 36 years of age said they did not feel the robot to be a human (or felt it little), while below that age, the responses were more distributed. The same distribution of answers was found in the question of whether they felt it was a living being. Thus, it seemed that the younger users, who showed a greater emotional response, tended to perceive the robot as a living being rather than a machine, compared to the older users.

Users’ responses to questions about their feelings towards the robot and its perceived empathy are presented in Figure 7. It can be seen that the majority of users responded neutrally or positively, indicating that they perceived the robot as friendly and capable of conveying empathy. Negative responses were mostly from users who appeared tense or uncomfortable during the session. It is noteworthy that, although users generally perceived the robot as friendly, some indicated in the first question that the robot’s tone of voice could suggest unfriendliness, which was compensated by the robot’s smooth conversation and supportive gestures. While initial indications suggested that users generally perceived the robot more as a mechanical entity rather than a living being, Figure 7 illustrates that the robot’s dialogues, gestures, and use of colours effectively conveyed a friendly and empathetic nature. As a result, users’ perceptions of the robot were positively influenced.

Figure 8 presents the users’ responses to questions regarding their overall feelings about the robot during the interview. The majority of users responded positively, expressing that they felt safe and comfortable while interacting with the robot. However, users who exhibited signs of tension and discomfort during the session, as seen in the recorded videos, evaluated their experience negatively.

Users’ responses to questions regarding the robot’s interaction during the interview are depicted in Figure 9. The majority of users perceived the robot as interactive, with no user selecting the option “not at all”. Unlike other question blocks, the distribution of answers varied in a clearly homogeneous and defined trend, as even users who felt tense or uncomfortable perceived the robot’s proximity or movements correctly or positively.

Figure 10 shows the median of the participants’ answers according to their age. As it was a small sample, the results of those aged 26 and over were grouped. There was a significant difference in the criteria between the two population groups, which corresponded to age, irrespective of gender. Young people responded in a more emotional way, while the older population had a more objective and measured answer.

In order to visualise and study the correlation between each pair of variables, the Pearson correlation coefficient was employed, as seen in Figure 11.

In this table, a high number of variables showed a significant dependency between them, and some of these relationships were expected, while others provided insightful information. The positive values confirmed that, as positive feelings towards the robots, such as friendliness and empathy, increased, positive feelings such as comfort and safety sensations also increased. Likewise, most negative characteristics, such as a fake or mechanical feeling, tended to change in the opposite direction as the more positive variables.

Nonetheless, the “feels unfriendly” variable appeared to move in the opposite direction as the other negative variables, having a positive correlation with other good feelings and characteristics. This may be due to the fact that there were few examples where the answer to the unfriendliness feeling was different from “not at all”, and even in these few cases, the participants continued to give positive answers in most other aspects. This situation could suggest that this variable is not that significant for people to have a good overall experience or that participants in these cases may not have been sure of the question and its meaning, which might cause the correlation measure to behave this way.

### 3.2. Qualitative Results

The qualitative data analysis process involved several steps. Initially, the transcribed material and videos of the sessions were reviewed, and an inductive thematic approach was used to identify content units, emerging themes, and patterns in the data. The researchers initially supplied an unrestricted text pertaining to each video, expressing their assessments of each participant in relation to nonverbal communication. This led to the establishment of initial categories, which were related in meaning, and each coded unit could have more than one category. This initial analysis was performed independently.

Successive meetings were held to establish the coding dynamics and discuss any differences in the excerpts selected and the categories chosen. The material was reviewed, and any discrepancies were resolved through analytical conversations until a consensus was reached regarding the excerpts, observational categories, and emerging themes.

Using a data reduction process, a “book of categories” was agreed upon, with a brief description of each one, which was later expanded during the process. From these initial dimensions, an inductive process was carried out, and the new categories are shown in Table 3 emerged.

In the second round of analysis, a deductive thematic approach was used, and the process was completed by saturating the information and developing new interpretations of meanings. The most-important quotes were extracted and grouped to organise all the information found, looking for correlations between them.

Finally, a narrative was written that connected the findings from the triangulation of all the data analysed. The categories are written in bold in the narrative.

The dimension called **environment** was also included, which was further divided into several categories.

In the **room** category, users expressed their focus on the environment, including factors such as temperature, lighting, and noise level. They indicated that the closed classroom setting, with air conditioning and no external noise, made them feel more comfortable, which contributed to a more successful session.

The **video camera** category reflects how users felt about being recorded, which was a mitigating factor in increasing the tension and stress of some users.

The **user body language** category was extrapolated from the visual analysis of the recordings, where participants’ gestures and expressions were examined. In the videos, it was observed that some users felt uncomfortable when they saw the video camera recording. However, some relaxation was perceived during the session’s development. Some users responded positively when the robot offered to take a selfie with them, but some students preferred the robot to take the photo itself.

The dimension called the **possibility of talking to a robot** can be divided into several categories.

The category of **selfie** reflects users’ willingness to take a selfie with the robot. The vast majority of users considered it a positive experience and welcomed the fact that the robot asked them to take a selfie. However, some users who were more tense and uncomfortable found it disturbing. Some users indicated that they would have preferred if the robot took the selfie itself instead of them using their own phone.

In the category of **movement**, some users noted that the robot’s gestures were only present in the questions and answers, giving it an artificial and unnatural feel. However, those users who were more comfortable with talking to the robot perceived the movements as natural, both attitudes reflecting the importance of the embodiment for better interaction. A group of users reflected that the robot had repetitive hand movements when it was static, which they found uncomfortable. For instance, User #10, who felt uncomfortable with the robotic platform, did not perceive it as human, but rather as completely mechanical and artificial. As a result, he did not feel completely safe during the session and did not develop or perceive much empathy for the robot.

The **technical aspects** category of the robot refers to its language, speech, eye colours, and conversation structure.

The analysis of the video recordings unveiled instances during certain sessions where the robot’s responses experienced delays or errors in **language processing**. These occurrences were found to have a significant impact on the participants, eliciting feelings of frustration and detachment. As a result, the participants faced difficulties in actively engaging in the remaining parts of the session.

Regarding the **conversation structure**, participants provided valuable feedback regarding their experiences interacting with the robot. In particular, many participants commented on the aspect of assigning colours to the robot’s eyes, which gave them the impression that the robot had a look that conveyed certain emotions.

For several participants, conversing with a robot was a new and interesting experience for them, which helped them maintain a positive attitude and willingness to engage with the robot. For instance, User #14 commented that interacting with the robot was a fascinating learning experience. Other users were more focused on the technical details of the session, such as User #8, who mentioned that he would have preferred a longer and more coherent conversation with the robot. Many users expressed their desire for a two-way conversation where they could ask questions of the robot as well. On the other hand, some users felt that the robot spoke too fast, making it difficult to understand at times.

These findings align with the patterns observed in the previously presented graphs, indicating that the embodiment of the robot had a positive impact on users’ emotions during the conversation. User #8, who commented on technical aspects and movement, perceived the platform as a robot, but did not feel empathy towards it. Nonetheless, he/she enjoyed the session and found the conversation to be pleasant.

A dimension called **interview** was included, with a category named **conversation**, where participants focused on their feelings during the communication and how the interaction unfolded. The value of co-interaction and communication with the robot during the session was positively highlighted by the participants. For example, User #1 commented, “The phrases and the rhythm are good”. Users also appreciated the fact that the robot took turns speaking and allowed them to speak. User #26 expressed the following: “I found it very interesting that the robot fully respected my turn to speak, and the answers I received were quite natural”.

Regarding the **interaction**, User #6 said, “I loved the way he laughed and the way he made a move towards me; it made me want to hug him”. User #26 initially felt uncomfortable, as he was not accustomed to talking to a robot, but as the chat progressed, he found it curious and amusing and felt comfortable throughout the interview. This response is in line with that of other users, such as User #29, who said, “I felt a bit strange at the beginning, but then I felt confident”. However, some users did not feel comfortable, such as User #10, who replied, “I felt quite uncomfortable at some point as you don’t feel like you are talking to a human person”. Several users noted that the moment when Pepper told them a joke had a positive impact on the session, as it helped to break the ice and establish a more relaxed and comfortable atmosphere. The video recordings confirmed that this moment played a crucial role. In fact, some participants explicitly stated that they found the joke to be amusing and it even made them laugh.

The analysis of the qualitative results led to the inference that the personality and behaviour of social robots play a crucial role in designing an improved user experience. It is noteworthy that all the positive experiences reported by the participants occurred when the robot demonstrated friendly behaviour and exhibited a personality that resembled human-like traits. These findings underscore the importance of integrating these qualities into the design of social robots to ensure a positive and engaging user experience. By considering the human-like aspects of personality and behaviour, designers can create social robots that establish a more positive and meaningful interaction with users.

No participants were found who expressed a desire to abstain from interacting with Pepper. This could be because they voluntarily chose to attend the session, indicating their willingness to engage with the robot.

## 4. Conclusions, Limitations, and Future Research

The aim of the study was to explore the user experience when communicating with a robot, considering the embodiment of the robot and verbal communication. Our evaluation included observing any behavioural modifications of participants during the session and analysing the results to determine if they met our initial objectives.

The study found that using a robotic platform for interaction had advantages and limitations, and there were lessons learned from each case.

The main contribution of this paper is the proposal of a method for endowing a social robot with the ability to maintain an open conversation, adaptable to the responses and moods of each user. This method is based on the use of modulation conversation profiles. In these profiles, developers can specify the effect of a particular emotion or mood of the user and modify the conversation to change the user´s mood or make him/her feel more comfortable. For each robot’s communicative interface, the profiles define a configuration for the interface’s parameters, such as sentences, gestures, colours, etc. This work established guidelines for replicating an experiment based on an informal conversation between a human being and a humanoid robot. This experiment can now be replicated in different environments and with different types of users, allowing for a larger sample of analysis and a richer extrapolation of the results.

The qualitative assessment of individual sessions consisted of one video for each participant. To conduct the evaluation, the authors carefully viewed all the videos twice. Initially, they offered descriptive comments for each video, outlining their findings concerning each participant’s nonverbal language, which were subsequently categorised. In a subsequent phase, a predefined set of criteria was employed to evaluate the videos. Any discrepancies were resolved through analytical discussions until a consensus was achieved.

From the triangulation of the qualitative and the quantitative results, it can be concluded that the participants generally enjoyed interacting with the robot, finding it novel and engaging. The video recording provided relevant information showing that the nonverbal communication and expression of each participant matched with the responses obtained in the forms.

Most users perceived the robot as friendly and capable of conveying empathy. However, delays or errors in the robot’s responses caused frustration and disconnection, which made it difficult for the participants to engage in the rest of the session. Even though the number of participants over 25 years old was small, the study found that it seemed that older participants were more objective in their views of the platform compared to younger participants, regardless of their gender.

The researchers observed that users predominantly perceived the robot as a mechanical entity, rather than a living being. However, the study showed that the robot’s dialogues, gestures, and use of colours effectively conveyed a friendly and empathetic disposition, influencing users to view it more positively. This underscores the importance of creating a comfortable conversational environment for users, where they can discern emotional elements in the robot’s behaviour, for example incorporating humour (jokes) accompanied by corresponding gestures, colours, and sounds of laughter or displaying genuine interest in the user. By integrating embodiment (including presence, body language, and colours) with well-crafted dialogues, gestures, and emotive sounds, users can develop increased confidence and comfort when interacting with the robot, fostering a sense of empathy for and from the robot.

In social robotics, the significance of user experience cannot be underestimated. The comfort and trust that users feel when interacting with robots are key. By designing robots considering embodiment, they become more relatable, and users are more likely to empathise and feel more comfortable during the interaction. Participants reported a more positive experience when the robot displayed friendly behaviour and a more human-like personality.

The recognition of the significance of personality and behaviour in social robots is essential for the design of an enhanced User Experience (UX). It was concluded that the integration of gestures, dialogues, and colours is crucial for enhancing the user experience with embodied robots. Gestures can make the interaction more natural and intuitive, while dialogues provide clear communication and guidance. Additionally, colours can effectively convey information and emotions. By combining these elements, the robot can transform into an engaging and trustworthy companion, thereby improving acceptance, usability, and trust during the interaction. Ultimately, user experience and embodiment are closely intertwined, and their synergy contributes to the success of various social robotics applications.

According to the study, customising the activities and responses of robots is crucial to adapt them to the unique characteristics of each user and to the specific applications and tasks of the robot. Enhancing language processing and emotion recognition algorithms using AI can promote more seamless and satisfactory communication for users.

In academic environments, the utilisation of embodied robots can enhance learning by providing a more engaging and interactive experience. In practical domains such as healthcare, embodied robots have the potential to perform tasks that instil a sense of security in users and convey empathy. Going forward, it is crucial to conduct additional research aimed at enhancing robot language processing algorithms and refining human–robot interaction protocols to foster a more inclusive relationship between robots and humans.

## Figures and Tables

**Figure 1 sensors-23-05274-f001:**
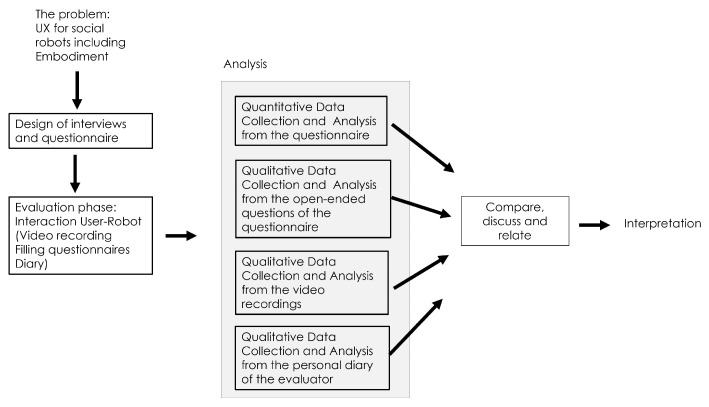
Methodology to evaluate the user experience.

**Figure 2 sensors-23-05274-f002:**
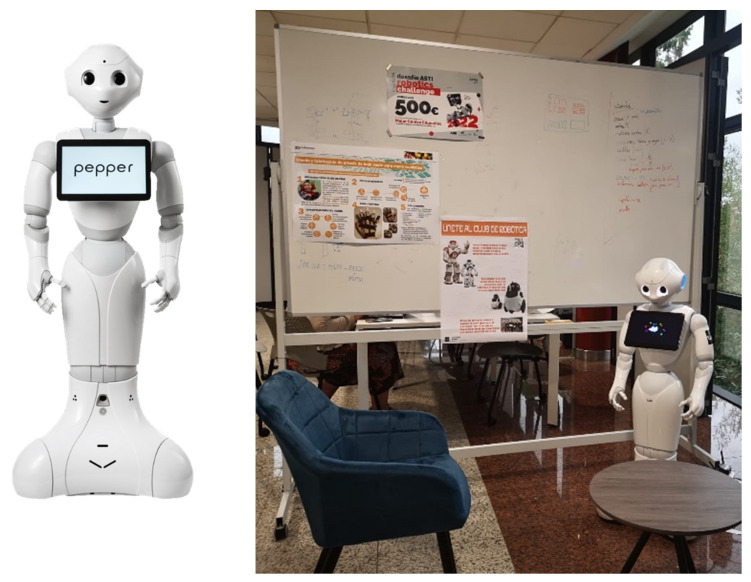
The Pepper robot and the environment of the experiment.

**Figure 4 sensors-23-05274-f004:**
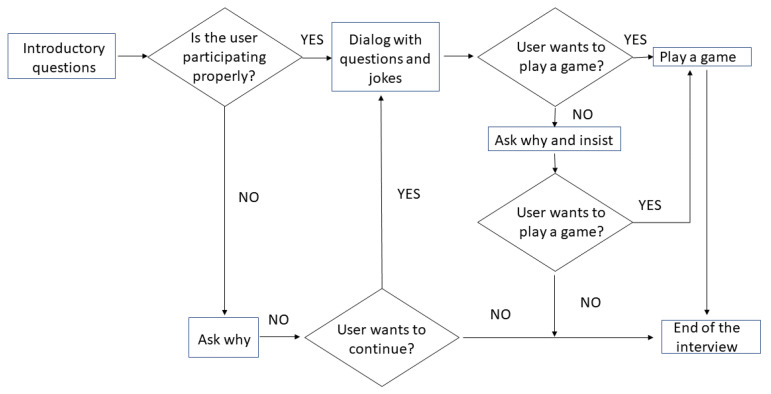
Flowchart diagram of the conversation.

**Figure 5 sensors-23-05274-f005:**
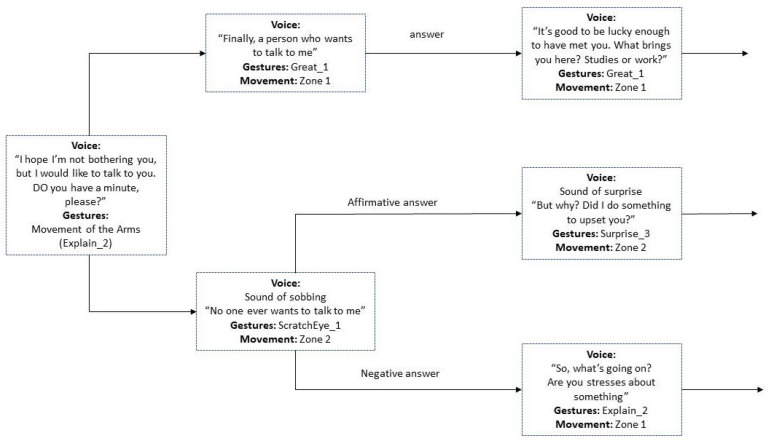
Diagram of the robot’s initial dialogue with the user, including voice and gestures.

**Figure 6 sensors-23-05274-f006:**
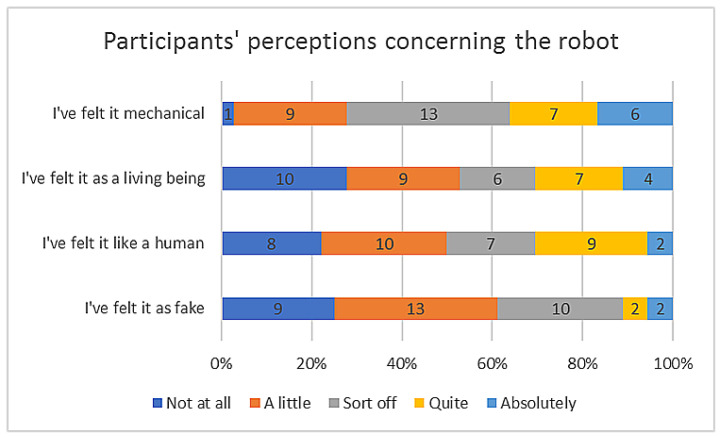
Participants’ perceptions concerning the robot.

**Figure 7 sensors-23-05274-f007:**
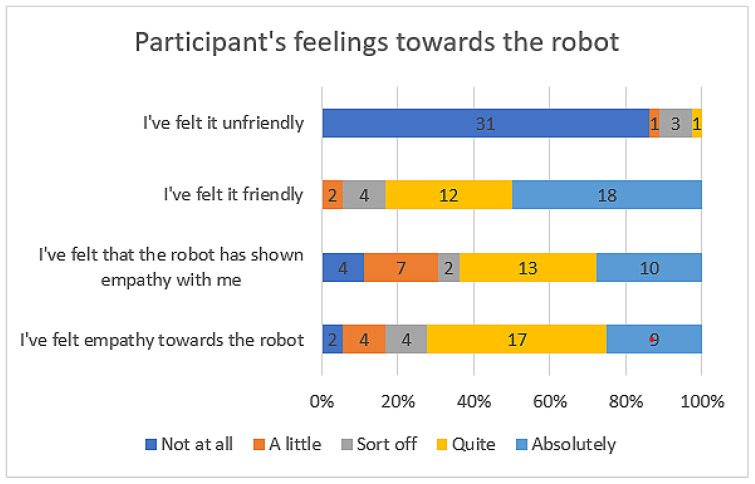
Participants’ feelings towards the robot.

**Figure 8 sensors-23-05274-f008:**
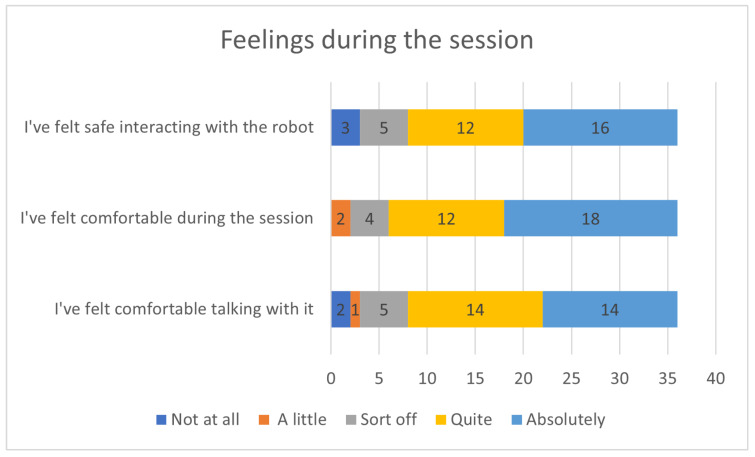
Feelings during the session.

**Figure 9 sensors-23-05274-f009:**
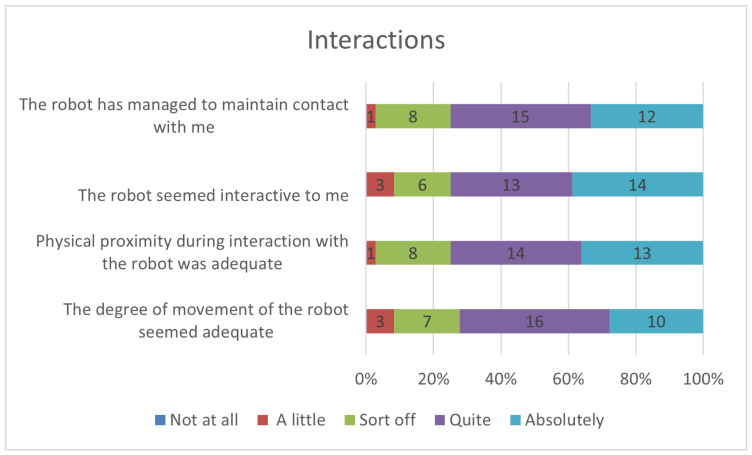
Perceptions of robot interactions.

**Figure 10 sensors-23-05274-f010:**
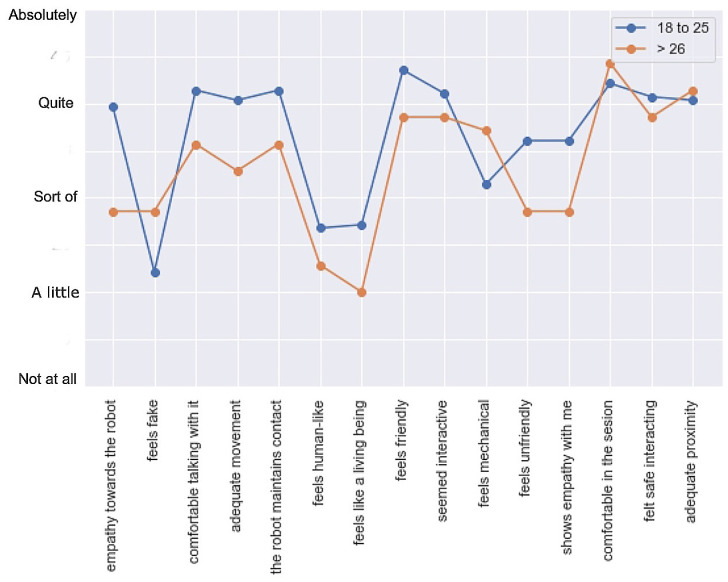
Median of answers by age.

**Figure 11 sensors-23-05274-f011:**
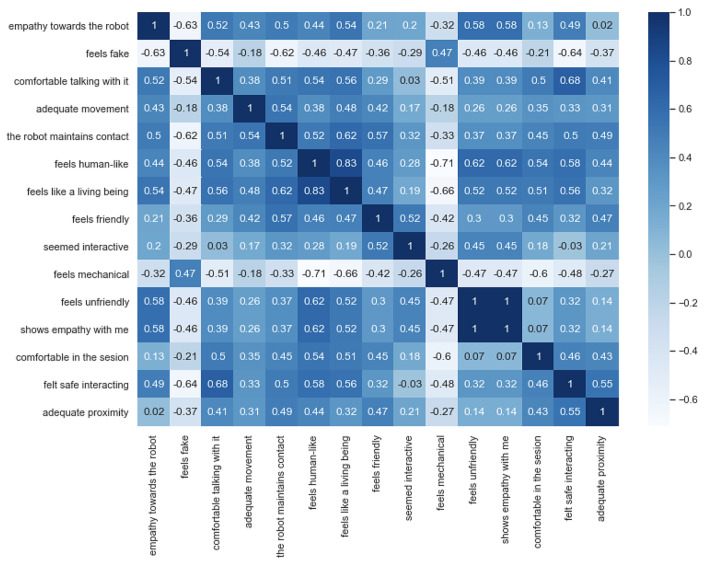
Pearson correlation coefficient of results.

**Table 1 sensors-23-05274-t001:** Robot movements with their description and the eye colour associated with them.

Name	Movements and Eye Colours
Great_1	It nods its head. The robot also raises its right arm to the level of its torso and brings its elbow backwards. Then, it lowers it slightly.
ScratchEye_1	It raises its arm towards its eyes and rotates it repeatedly (as if pretending to scratch its eye). Eyes with light blue and blue LEDs.
Surprise_3	It raises its head, looking forward. It opens its arms slightly, palms facing outwards. Gasp of surprise. Eyes with yellow LEDs.
Shocked_1	It looks straight ahead. It moves its torso forward, crouching slightly. It brings its hands close to each other and separates them, leaving its arms hanging by its sides. Eyes with blue LEDs.
Mocker_1	It looks up, then lowers its head to the side and repeatedly raises and lowers it. The robot points forward with its left arm. It moves its right arm into a fist towards the lower part of the torso. Mocking laugh sound. Eyes with white LEDs.
Laugh_2	It lowers its head and then raises it, looking straight ahead. It bends its right arm putting its fist on its chin. It bends its left arm bringing the fist to the lower part of the torso. Sound of brief discreet laughter. Eyes with light blue and green LEDs.
Happy_4	It shakes its head up and down as if nodding. It bends its arms up slightly, bringing its hands closer. Eyes with green LEDs.

**Table 2 sensors-23-05274-t002:** Age of the participants.

Answer	Age
18 to 20 years	47.22%
20 to 25 years	30.56%
26 to 30 years	2.78%
30 to 40 years	5.56%
more than 40 years	13.89%

**Table 3 sensors-23-05274-t003:** Categories that emerged from the qualitative analysis.

Dimensions	Categories	Sub-Categories
Environment	Room	
	Video camera	
	User body language	
Talk to a robot	Selfie	Positive
		Disturbing
		Auto selfie
	Movement	Natural
		Only in answer
		Move hands
	Technical aspects	Language processing
		Structure
Interview	Conversation	
	Interaction	

## Data Availability

The original data of this contribution can be requested from the corresponding author.

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
