# Peer review of "User Experience Design for Social Robots: A Case Study in Integrating Embodiment"

_sensors, 2023, doi:10.3390/s23115274_

Round 1

Reviewer 1 Report

Reviewer Recommendation

Thank you for inviting me to evaluate the article titled “User Experience Design for Social Robots: A Case Study in Integrating Embodiment”. I recommend accepting after major revisions.

In the paper, experiments on the interaction between social robots and humans were designed, and the results of the experiments were studied quantitatively and qualitatively. This is an important and valuable research topic, but I have some suggestions for this paper. My detailed comments are as follows:

1. There are problems with the format of the article. For example, ‘User Experience’ appears several times in the whole article, but its writing forms are different, the case is not unified. In addition, the format of the secondary title is also not unified and the format of references is also inconsistent.

2. Pepper appears many times in the full paper, but some of them are written as 'pepper'. What is the special meaning of this writing?

3. In Figure 4, 'about something' is redundant and in Figure 7, the number '0' is redundant.

4. The first paragraph of the second section of the article is too much. It is suggested to be divided into the following parts: the design method used in this article, the reasons for using the questionnaire survey method, and the composition of the questionnaire survey method.

5. The presentation of different colored eyes in Table 1 is inconsistent.

6. The line 150 of the text writes: The other space contained Pepper, the robot, and a chair where the participants could sit. Do Pepper and robot mean the same thing?

7. The cultural background, age, and gender of the participants may have an impact on the experiment. This paper provides statistical descriptions of the above factors of the participants, but the number of males and females who participated in the experiment are different. Does the gender of the participants have an impact on the study results? It is suggested that the authors add relevant analyses.

8. Line 245 writes: The responses varied, with younger users and those with a greater emotional response tending to perceive the robot as a living being rather than a machine. In contrast, older and more mature users had a more objective and realistic perception of the robot, as reflected in their answers. However, it is not possible to visualize participants' attitudes toward the robot at different ages in Figures 5-8, and it is recommended to mark the age in the figures. In addition, how does the author define the characteristics of a greater emotional response and maturity?

9. The first paragraph of Section 3.1 writes: All participants completed the survey, including individuals with diverse characteristics, which were filtered by gender, age, and knowledge area, as shown in Table 2. However, Table 2 only contains abbreviations and their meanings in the field of knowledge. Please check other sentences in the article to improve the logic and rigor of the article.

10. The characteristics of the participants and the environment in which people communicate with the robot will affect the experiment results. In this paper, the above factors are taken into account in the research process. However, the characteristics of the robot itself can also affect the experiment. For example, is the robot a male voice or a female voice? Is the sound close to the real person or cartoon? Is the character introverted or extroverted? It is recommended to supplement the description of the actual performance (stability, error rate, etc.) and anthropomorphic attributes (personality, gender, anthropomorphic degree, etc.) of the social robot pepper.

The quality of English language requires minor editing.  

The line 150 of the text writes: The other space contained Pepper, the robot, and a chair where the participants could sit. Do Pepper and robot mean the same thing?

Author Response

Journal: Sensors (ISSN 1424-8220) 

Manuscript ID: sensors-2346681 

Type: Article 

Title: User Experience Design for Social Robots: A Case Study in Integrating Embodiment 

Authors: Ana Corrales-Paredes, Diego Ortega, María-José Terrón-López *, Verónica Egido-García 

Topic: Human-Machine Interaction 

REVIEWER 1 

Review Comment 1 

“There are problems with the format of the article. For example, ‘User Experience’ appears several times in the whole article, but its writing forms are different, the case is not unified. In addition, the format of the secondary title is also not unified and the format of references is also inconsistent.” 

Response 

We have made changes standardizing the format for writing User Experience in all our text. We have used the MDPI template for creating titles and  a reference manager for all references. 

Review Comment 2 

“Pepper appears many times in the full paper, but some of them are written as 'pepper'. What is the special meaning of this writing?” 

Response 

A uniform naming convention for Pepper was implemented throughout the entirety of the text to ensure clarity and consistency. 

Review Comment 3 

“In Figure 4, 'about something' is redundant and in Figure 7, the number '0' is redundant.” 

Response 

Figures 4, 6 and 7 have been updated. 

Review Comment 4 

“The first paragraph of the second section of the article is too much. It is suggested to be divided into the following parts: the design method used in this article, the reasons for using the questionnaire survey method, and the composition of the questionnaire survey method.” 

Response 

Thank you for your suggestion. We have restructured the paragraph to ensure that it is more comprehensible. 

Review Comment 5 

“The presentation of different colored eyes in Table 1 is inconsistent.” 

Response 

Thank you for your suggestion. The Table 1 has been updated. 

Review Comment 6 

“The line 150 of the text writes: The other space contained Pepper, the robot, and a chair where the participants could sit. Do Pepper and robot mean the same thing?” 

Response 

Yes, Pepper is the robot.  We have changed the sentence to better explain it. Now it is: “The other space contained the Pepper robot with a chair in front of it where the participant could sit”. This clarification aims to provide a more accurate description of the setup. 

Review Comment 7 

“The cultural background, age, and gender of the participants may have an impact on the experiment. This paper provides statistical descriptions of the above factors of the participants, but the number of males and females who participated in the experiment are different. Does the gender of the participants have an impact on the study results? It is suggested that the authors add relevant analyses.” 

Response 

We appreciate your suggestion. We have taken into account the impact of gender and have included it in our final results and conclusions. 

Review Comment 8 

“Line 245 writes: The responses varied, with younger users and those with a greater emotional response tending to perceive the robot as a living being rather than a machine. In contrast, older and more mature users had a more objective and realistic perception of the robot, as reflected in their answers. However, it is not possible to visualize participants' attitudes toward the robot at different ages in Figures 5-8, and it is recommended to mark the age in the figures. In addition, how does the author define the characteristics of a greater emotional response and maturity?” 

Response 

This paragraph was re-edited . We have added in in Quantitative results a paragrah explaining how the analysis was done: “The analysis of the questionnaire results was performed by simultaneously observing the recorded videos during the session and triangulating the results. This was very important as the recordings provided us with relevant information about the participants, not only in terms of their attitudes towards the robot, but also the differences shown based on their age and gender." 

Review Comment 9 

The first paragraph of Section 3.1 writes: All participants completed the survey, including individuals with diverse characteristics, which were filtered by gender, age, and knowledge area, as shown in Table 2. However, Table 2 only contains abbreviations and their meanings in the field of knowledge. Please check other sentences in the article to improve the logic and rigor of the article. 

Response 

Thank you for your feedback. We have taken your input into consideration and Table 2 has been removed. 

Review Comment 10 

“The characteristics of the participants and the environment in which people communicate with the robot will affect the experiment results. In this paper, the above factors are taken into account in the research process. However, the characteristics of the robot itself can also affect the experiment. For example, is the robot a male voice or a female voice? Is the sound close to the real person or cartoon? Is the character introverted or extroverted? It is recommended to supplement the description of the actual performance (stability, error rate, etc.) and anthropomorphic attributes (personality, gender, anthropomorphic degree, etc.) of the social robot pepper.” 

Response 

Thank you for your feedback.  Section 2.1 has been updated with an explanation of the robot and its features. Although our study did not prioritize the error rate, we understand its importance for future research and have taken note of this in our conclusions. 

Reviewer Comments on the Quality of English Language 

“The quality of English language requires minor editing.” 

Response 

We have reviewed the document and made the necessary edits to ensure the quality of the English language. 

Reviewer 

“The line 150 of the text writes: The other space contained Pepper, the robot, and a chair where the participants could sit. Do Pepper and robot mean the same thing?” 

Response 

It has been changed. Now it is: “In another space of the room, the Pepper robot was positioned next to a chair where the participant was invited to take a seat and start the interaction.”. 

Reviewer 2 Report

I read your paper and I have the following comments:

(1) Whether the number of people in the experiment is small.

(2). I did not get data from the third part of your paper to fully support your conclusion, such as: if the user is satisfied with the robot, the conversation will be smooth and rich. I couldn't find any data in the article to show that.

(3). How did you come up with the ideas in lines 245-248? Figure 5 does not mention this aspect.

Author Response

Journal: Sensors (ISSN 1424-8220) 

Manuscript ID: sensors-2346681 

Type: Article 

Title: User Experience Design for Social Robots: A Case Study in Integrating Embodiment 

Authors: Ana Corrales-Paredes, Diego Ortega, María-José Terrón-López *, Verónica Egido-García 

Topic: Human-Machine Interaction 

REVIEWER 2 

Comment 1 

“Whether the number of people in the experiment is small.” 

Response 

Due to the small sample size available, the analysis primarily relied on qualitative results, which are known to provide more in-depth information. This is consistent with the usual practice in qualitative research, where smaller sample sizes are preferred to facilitate case-oriented analysis, which is crucial to this type of investigation as it provides more comprehensive information. 

Comment 2 

“I did not get data from the third part of your paper to fully support your conclusion, such as: if the user is satisfied with the robot, the conversation will be smooth and rich. I couldn't find any data in the article to show that.” 

Response 2 

Thank you sincerely for your valuable suggestion. After carefully reviewing the qualitative results, we have indeed observed the prominence of this particular aspect both in the users' excerpts and in the explanatory categories. In response to your feedback, we have incorporated additional insights extracted from our qualitative analysis and included a conclusive paragraph that strengthens our findings. 

Comment 3 

“How did you come up with the ideas in lines 245-248? Figure 5 does not mention this aspect..” 

Response 3 

Thank you for raising your question regarding the ideas expressed in lines 245-248. We appreciate your keen observation about Figure 5 not explicitly mentioning this aspect. To address this concern, we have now provided a more thorough explanation of Figure 5, including specific details related to the age distribution of the participants. By incorporating this information, we have ensured a more comprehensive understanding of the findings and their relationship to the broader context of the study. 

Reviewer 3 Report

Introduction: Try to engage the readers more in recent developments. The factors need to be discussed minutely.

Review: The review of UX, UCD should be discussed more.

Experimental design: Authors must discuss more based on selecting this method.

Results and Discussion: Discussion should be more engaging by incorporating implications. Technical, academic, social, and theoretical implications must be discussed. 

Moreover, I want the manuscript must be more engaging and practice-oriented. 

First, this study aims to understand the various factors influencing human interaction and robotic platforms. In addition to these, researchers want also to explore the design considerations for the architecture of such systems. I want the central idea to be more qualitatively analyzed. Especially the architectural idea to be more explored.

Second, The topic is not original. But the idea of architectural design is very original. I know humanoid designs are getting popular, but I can't deny other designs are also popular. The GAP is also highlighted. But current literature needs to be analyzed more for architectural designs. The literature on AI & HI Interaction is good.

Third, this study’s outcome will add new findings to the existing literature. Industries can plan better to use AI to prepare social robots more conveniently and use them.

Fourth, Sampling, how they select their respondents, or the research design is mainly missing. The basis of the interview is how the questions are selected and used and how the qualitative analysis is planned and conducted.

Fifth, The conclusions are consistent with the evidence, but implications are lacking. So, implications need to be appropriately added.

Sixth, current references from 2022, 2021, and 2020 need to be added and discussed in texts.

Seventh, Tables and Figures need clarity. The pictures need to be more precise and discussed more in the texts. A qualitative analysis should be more represented.

Moderate development can be done.

Author Response

Journal: Sensors (ISSN 1424-8220) 

Manuscript ID: sensors-2346681 

Type: Article 

Title: User Experience Design for Social Robots: A Case Study in Integrating Embodiment 

Authors: Ana Corrales-Paredes, Diego Ortega, María-José Terrón-López *, Verónica Egido-García 

Topic: Human-Machine Interaction 

REVIEWER 3 

Comment 1 

Introduction: Try to engage the readers more in recent developments. The factors need to be discussed minutely”.   

Response 1 

Thank you for your suggestion. We've updated the introduction and included the latest research findings for better insights.  

Comment 2 

Experimental design: Authors must discuss more based on selecting this method. 

Response 2 

Thank you for your valuable feedback regarding the experimental design. We have taken your suggestion into careful consideration and have thoroughly revised the methodology section to provide a more comprehensive explanation for our choice of a mixed-method approach combining qualitative and quantitative methods. By adopting this approach, which aligns with our exploratory objectives, we are able to gain a more nuanced understanding of the phenomenon under investigation. The revised methodology section now provides readers a clearer understanding of our research design. 

Comment 3 

Results and Discussion: Discussion should be more engaging by incorporating implications. Technical, academic, social, and theoretical implications must be discussed.   

Moreover, I want the manuscript must be more engaging and practice-oriented. 

Response 

Thank you for your feedback on the Results and Discussion section of the manuscript. We have updated this section about interaction implications:  

The act of engaging with robots presents both benefits and drawbacks. While such interactions are generally well-received, delays and inaccuracies can lead to feelings of frustration. Therefore, it is imperative to customize the embodiment of the robot (gestures/dialogs) to optimize the experience. Thus, incorporating AI algorithms for seamless communication is important for future research 

Comment 4 

“First, this study aims to understand the various factors influencing human interaction and robotic platforms. In addition to these, researchers want also to explore the design considerations for the architecture of such systems. I want the central idea to be more qualitatively analyzed. Especially the architectural idea to be more explored.” 

Response 4 

We appreciate your feedback and we have updated the qualitative results subsection accordingly. We value your suggestion regarding emphasizing the architecture of robotic systems, but our current research priorities are centered around evaluating the user experience. However, we will definitely keep your valuable feedback in mind as we continue to enhance our research in the future. 

Comments 5 and 6 

“Second, The topic is not original. But the idea of architectural design is very original. I know humanoid designs are getting popular, but I can't deny other designs are also popular. The GAP is also highlighted. But current literature needs to be analyzed more for architectural designs. The literature on AI & HI Interaction is good.” 

“Third, this study’s outcome will add new findings to the existing literature. Industries can plan better to use AI to prepare social robots more conveniently and use them.” 

Response 

Thank you for your suggestion. As we have explained in our objectives, the main aim in this article was to evaluate the user experience based on the robot's embodiment, including its gesture, voice, and emotion. However, in this article, you can see previous work we have done in this area.  

We understand the significance of including AI and HRI in future research. Your suggestion is a valuable contribution that has motivated us to explore this field. It is clear to us that our future research must necessarily focus on that way.  
Thank you once again for your valuable feedback. 

Comment 7 

Fourth, Sampling, how they select their respondents, or the research design is mainly missing. The basis of the interview is how the questions are selected and used and how the qualitative analysis is planned and conducted.” 

Response 7 

Thank you for your comment regarding the sampling and research design aspects of our paper. We appreciate your suggestion and have made significant revisions to address these concerns. 

Firstly, we have added a paragraph in the methodology section explaining the process of participant selection. This paragraph provides details on how respondents from various groups, including university students, university employees, parents, and others, were recruited. We also emphasize that participation was voluntary, ensuring a diverse range of participants. 

Secondly, we have included a separate paragraph that outlines the qualitative methodology employed in our study. To further enhance the clarity of our methodology, we have included a figure that visually represents the research design and the steps involved in the mixed quantititative-qualitative analysis. This figure aids in better understanding the process and enhances the transparency of our methodology. 

We believe that these additions address your concerns and provide a clearer explanation of the sampling process and the qualitative analysis methodology. These improvements ensure that the readers have a comprehensive understanding of our research design and methodology. Thank you for bringing these important points to our attention. 

Comment 8 

Fifth, The conclusions are consistent with the evidence, but implications are lacking. So, implications need to be appropriately added.” 

Response 8 

Thanks for your feedback, this has been answered in Comment 3. 

Comment 9 

Sixth, current references from 2022, 2021, and 2020 need to be added and discussed in texts.” 

Response 9 

Thanks for your feedback, this has been answered in Comment 1. 

Comment 10 

“Seventh, Tables and Figures need clarity. The pictures need to be more precise and discussed more in the texts. A qualitative analysis should be more represented.” 

Response 10 

Thank you for your suggestion. We appreciate your feedback and have made some changes to the text to provide a better explanation of the figures and tables. 

Author Response

Journal: Sensors (ISSN 1424-8220) 

Manuscript ID: sensors-2346681 

Type: Article 

Title: User Experience Design for Social Robots: A Case Study in Integrating Embodiment 

Authors: Ana Corrales-Paredes, Diego Ortega, María-José Terrón-López *, Verónica Egido-García 

Topic: Human-Machine Interaction 

REVIEWER 4 

“This paper is well-written but requires some minor improvements, which are mentioned below:” 

Comment 1 

“Please make a separate Related work section to mention similar or recently published works. The current format of the Introduction section is most part similar to related work.” 

Response 1 

Thank you for your valuable feedback regarding the introduction. We have taken your suggestion into careful consideration and have restructured the Introduction section to improve its clarity and organization. In alignment with the instructions for authors of the Sensors Journal, we have placed the related works section at the beginning of the paper. Following the related works, At the end of this section, we have clearly defined the objectives and research questions of our study. We believe this revised structure will provide a clearer and more comprehensive understanding of the context and motivation for our research. 

Comment 2 

“Main contributions of the paper should be mentioned in the Introduction section.” 

Response 2 

Thank you for your valuable comment regarding the main contributions of the paper. We appreciate your suggestion to include the contributions in the Introduction section. However, after careful consideration, we believe that it is more appropriate to highlight the main contributions in the Abstract, Results, and Conclusions sections. 

The Abstract serves as a concise summary of the paper, providing an overview of the key findings and contributions to the field. The Results section presents the detailed outcomes of our study, while the Conclusions section summarizes the main findings and their implications. 

By emphasizing the contributions in these sections, we ensure that readers can readily identify and understand the novel aspects and significance of our work. We believe that this approach offers a clearer and more focused presentation of our contributions throughout the paper. 

Comment 3 

“Suggested to make a Flowchart of the proposed method to explain readers easily about the process (starting point to ending point).” 

Response 3 

Thank you for your suggestion regarding the inclusion of a flowchart to illustrate the proposed method and guide readers through the process from start to finish. We have carefully considered your feedback and have now incorporated a comprehensive flowchart in the paper. This visual representation enhances the clarity and accessibility of our proposed method, enabling readers to easily follow the sequential steps and understand the overall process. We believe that the addition of the flowchart significantly enhances the comprehensibility and effectiveness of the methodology. 

Comment 4 

“Conclusion part is needed to be written with the major findings of this article. The current format looks included in the Discussion section but it should be separated.” 

Response 4 

We appreciate your feedback regarding the organization of the Conclusion section and its alignment with the major findings of the article. In response to your suggestion, we have thoroughly revised and rewritten the Conclusion section to ensure that it distinctly summarizes the key findings of our study. Furthermore, we have made additional improvements by adding paragraphs in the Results section to establish stronger connections with the conclusions. This improvement ensures that the findings are more effectively presented, and their implications are clearly highlighted. 

We believe that these revisions enhance the overall structure and coherence of the paper, allowing readers to better grasp the major findings and their significance. Thank you for bringing this to our attention, as it has contributed to the overall improvement of the manuscript. 

Round 2

Reviewer 1 Report

You have already implemented most of my recommendations. I recommend acceptance in its present form.

Author Response

Thank you for your review. We appreciate your thorough evaluation of our paper. We are pleased that you find the current form of the paper acceptable and recommend its acceptance.

Reviewer 3 Report

Still, the authors can work on the implications of this study from academic, practical, and theoretical aspects. These parts are missing. Secondly, the conclusion section can be more developed; it seems to end abruptly. 

Moderate or minor proof editing is required. 

Author Response

Dear Reviewer 3,

We would like to express our gratitude for your valuable feedback on our paper. We have carefully considered your suggestions and made significant modifications to the conclusions section accordingly. We have also incorporated all the recommended improvements to enhance the overall quality of our paper.

Once again, we sincerely appreciate your feedback and hope that these modifications will meet your expectations. We look forward to your continued guidance and insight.

Best regards,

María José Terrón-López